# Equivalent Dynamic Analysis of a Cable-Driven Snake Arm Maintainer

Guodong Qin [1,2], Huapeng Wu [1] and Aihong Ji [2],*

1   Laboratory of Intelligent Machines, School of Energy Systems, Lappeenranta-Lahti University of Technology, 53850 Lappeenranta, Finland; guodong.qin@student.lut.fi or guodongqin@nuaa.edu.cn (G.Q.); huapeng.wu@lut.fi (H.W.)
2   Laboratory of Locomotion Bioinspiration and Intelligent Robots, College of Mechanical and Electrical Engineering, Nanjing University of Aeronautics & Astronautics, Nanjing 210016, China
*   Correspondence: jiaihong@nuaa.edu.cn; Tel.: +86-138-5155-6844

**Abstract:** In this paper, we investigate a design method for a cable-driven snake arm maintainer (SAM) and its dynamics modelling. A SAM can provide redundant degrees of freedom and high structural stiffness, as well as high load capacity and a simplified structure ideal for various narrow and extreme working environments, such as nuclear power plants. However, their serial-parallel configuration and cable drive system make the dynamics of a SAM strongly coupled, which is not conducive to accurate control. In this paper, we propose an equivalent dynamics modelling method for the strongly coupled dynamic characteristics of each joint cable. The cable traction dynamics are forcibly decoupled using force analysis and joint torque equivalent transformation. Then, the forcibly equivalent dynamic model is obtained based on traditional series robot dynamic modelling methods (Lagrangian method, etc.). To verify the correctness of the equivalent dynamics, a simple model-based controller is established. In addition, a SAM prototype is produced to collect joint angles and cable forces at different trajectories. Finally, the results of the equivalent dynamics control simulation and the prototype tests demonstrate the validity of the SAM structural design and the equivalent dynamics model.

**Keywords:** snake arm maintainer; cable-driven; equivalent dynamic; special environment application

## 1. Introduction

A cable-driven snake arm maintainer (SAM) drive unit is generally placed outside of a robot's working space. This placement reduces the number of electronic devices on the entire snake arm, which is highly conducive to remote maintenance tasks in complex and narrow environments, such as nuclear power plants [1–3]. The China Fusion Engineering Test Reactor (CFETR) is large-scale, international scientific project adopting a remote handing system (RH) for regular monitoring and maintenance [4]. As shown in Figure 1, the SAM is installed at the end of the CFETR multipurpose overload robot (CMOR) and is transported to the inside of the vacuum chamber via a CASK transfer vehicle for reactor maintenance, flaw detection, and dust removal, as well as general monitoring and observation [5]. The SAM has a large enough dexterous operating space and is sufficiently adaptable in a high-radiation environment. It can remove a large amount of dust left by high-temperature plasma bombardment at the base of the divertor during operation of the test reactor. It can also obtain high-quality plasma and improve the operational stability of the test reactor. Traditional industrial robots cannot meet the requirements of narrow, multiobstacle, and high-radiation operations in a vacuum chamber environment. The SAM has a high slenderness ratio, and its redundant degrees of freedom allow for complete streamlining of motion, with strong environmental adaptability and obstacle avoidance ability. Therefore, the application potential for monitoring and maintenance of large, complex equipment in narrow spaces is extensive [6,7]. There are two main types of

SAM: underactuated (continuum) and rigid hyper-redundant arms. The majority of SAMs are formed in series using several basic joint units (flexible rods, springs, universal joints, etc.) [8–10]. OC Robotics initially defined a rigid hyper-redundant SAM and designed the snake arm using the principle of bionics before applying it to extreme environments, such as nuclear power plants [11]. Peng et al. [12] introduced a continuous SAM integrated with multilayer flexible plane springs that can realize precise linear movement. The authors of [13,14] designed a coupled SAM control scheme combining interactive obstacle control and path-following algorithms. The authors of [15,16] used the exponential integration method to solve the inverse kinematics of a hyper-redundant SAM. SAM has the characteristics of a series-parallel structure, and it is difficult to analyze the dynamics of its motion process. The dynamic equations established by the traditional Newton–Euler method, the Lagrangian method, the Gaussian method, etc., cannot clearly describe the relationship between the motion of each joint of a SAM and the force of the cable [17,18]. In addition, the motors, cables, joints, and end effectors of SAM involve complex mapping relationships, all of which have a coupling effect on the cable traction forces. As a result, the dynamic modelling of SAM and the solution of cable traction force in the movement process become increasingly complicated.

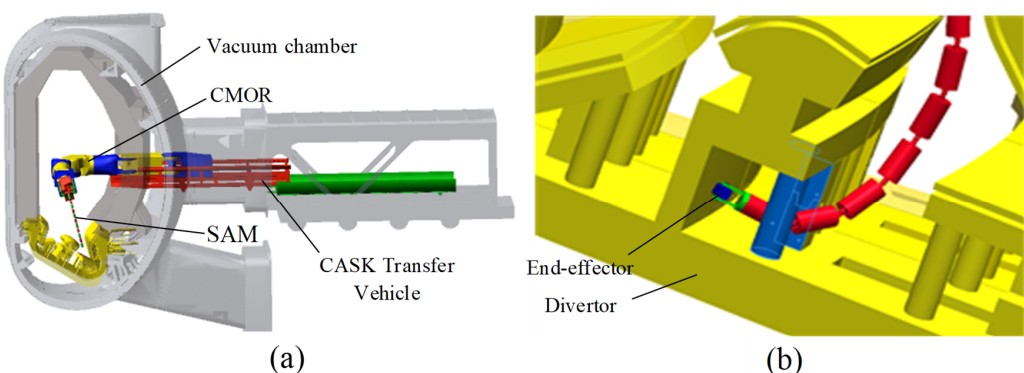

(a)　　　　　　　　　　　　　　　　(b)

**Figure 1.** SAM operation process. (**a**) SAM installs the end of the CMOR into the vacuum chamber; (**b**) maintenance inspection of the divertor.

Currently, there are two main methods for modelling the dynamics of a continuum SAM: the piecewise constant curvature method and the variable curvature method [19–21]. The former involves solving the dynamic equation of the entire arm by assuming that the deformation angle between each adjacent continuum is approximately equal (constant curvature) [22]. The latter involves performing dynamic modelling compensation of SAM using various compensation algorithms under the influence of gravity and load to improve the calculation accuracy [23]. Whereas the dynamic equation of a hyper-redundant SAM is similar to that of a rigid link robot, the configuration of the hyper-redundant series-parallel connection and the coupling effect of the drive cable increase the complexity of dynamic modelling [24,25]. Ciurezu-Gherghe [26] used the direct method and the superpositioning method to analyze the mode and dynamics of SAM to determine its vibration shape and natural frequency. Meanwhile, Vossoughi et al. [27] adopted the Gibbs–Appell method to obtain the dynamic equations of the in-plane motion and simplify the dynamics of SAM. Peng et al. [28] used a multiobjective particle swarm optimization method to simultaneously optimize the energy and control accuracy indicators in the SAM trajectory-tracking process. Xu et al. [29] proposed a dynamic control strategy using multilevel mapping for motion analysis and compensation. They calculated the feedforward torque of the SAM motor using recursive dynamics and the "rope pull-motor torque" relationship. Hua et al. [30] investigated the effect of different parameters on the vibration of a dynamic model by controlling for variables. Qin et al. [31] proposed a virtual force feedback algorithm to improve SAMs' perception and assistance capabilities in remote operations. Ma et al. [32] applied the perturbation method to a dynamic model of SAM, derived a vibration control

equation, and studied the natural frequency value of the cable when it was elastic. Due to the strong coupling between the joints, the end effectors, and the drive motors of the SAM, it can prove difficult to obtain the attendant mapping relationships directly. As such, it is difficult to establish an accurate dynamic model to calculate the driving force of each drive cable in real time, especially when the arm length and degree of freedom increase.

In this paper, an underactuated SAM applied to the narrow vacuum chamber of the CFETR is proposed. The kinematics equation of the SAM is subsequently established using the Denavit–Hartenberg (D–H) method, and the static cable driving force is calculated. For the strong coupling characteristics of SAM dynamics, an equivalent dynamics modelling method is proposed. The forcibly equivalent method is adopted for the transformation of joint torque and cable traction force. Then, the equivalent dynamic model can be obtained based on traditional series robot dynamics modelling methods (Lagrangian method, etc.). Finally, we establish a SAM dynamics controller and prototypes and verify the effectiveness of equivalent dynamics modelling through simulation and experiments.

The remainder of the paper is organized as follows. In Section 2, we introduce the underactuated SAM platform. In Section 3, we outline the equivalent dynamic analysis method. In Section 4, we introduce an equivalent dynamics controller. In Section 5, we present the related simulation and experiments of the single-joint group and a SAM with an arm length of 1500 mm. In the last section, we summarize the full text and provide several conclusions.

## 2. Related Work

### 2.1. Underactuated SAM Platform

As shown in Figure 2, the designed SAM arm is 2300 mm long and contains 10 joints. To suit the narrow environment, the underactuated principle is used for a lightweight and miniaturized design. The SAM designed for the actual needs of the CFETR vacuum chamber includes three main parts: a multijoint snake arm, an end effector, and a cable drive box. The movement of each joint involves the use of three evenly distributed cables along the circumference to achieve a two-degrees-of-freedom rotational movement. The end effector can be equipped with various tools, such as vision, clamping, cutting, and dust-removal tools, whereas the signal transmission is realized through the internal circular hole wiring of the joint unit. Meanwhile, multiple sets of servo motors are installed in the cable drive box, with the screw rod module driven by the servo motors to pull the cables and realize motion control of each joint group of the SAM.

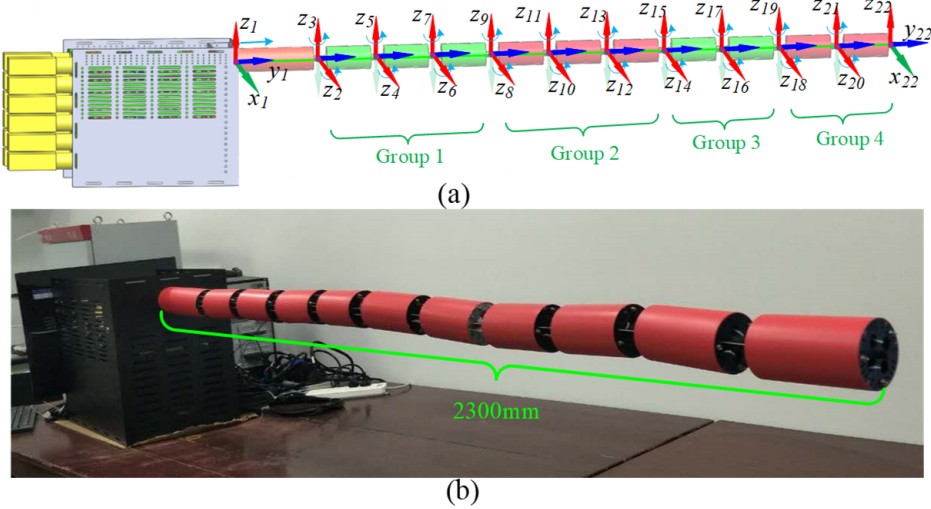

**Figure 2.** SAM design model (**a**) and prototype (**b**).

The principle behind the underactuated design lies in using a single-drive motor group (three motors) to drive multiple joint units to form a joint group to achieve synchronous rotation of the angle. Figure 3a shows the cable distribution of each joint in a single-joint group, where each joint cross-section contains 15 rows of cable through-holes. The first joint group contains three joints driven by cables in three rows of through-holes, denoted by the blue dashed lines in Figure 3a. The distribution of cables along the joint section is shown in Figure 3b. Here, each row of cables is fixed in layers in the through-holes of the first, second, and third joint units in the first joint group from the outside to the inside in the radial direction. The other end of the cable is driven by the screw rod module, as shown in Figure 3c. A transmission gear is installed at the end of each screw rod to realize the movement of the three screw rods in the screw-rod module in a ratio of 1 : 2 : 3. The screw-rod module drives the multiple joints in a joint group to realize the synchronous movement of the adjacent joints in the space.

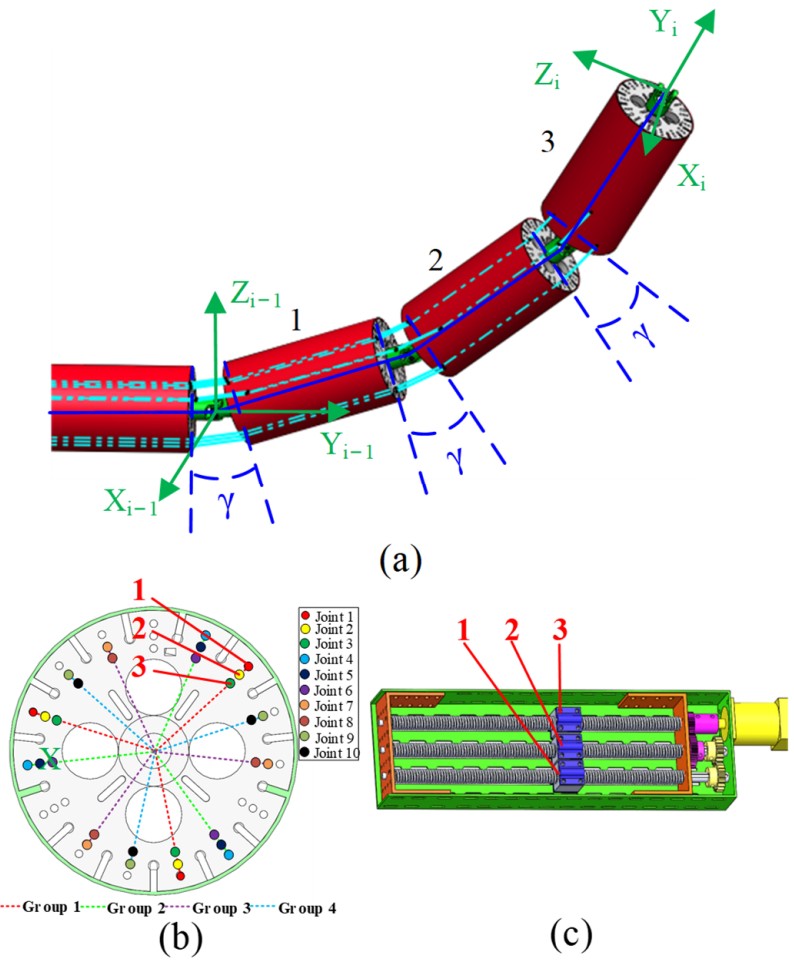

**Figure 3.** SAM underactuated mechanism and cable distribution. (**a**) Synchronous movement of the underactuated joint group rotation angle. (**b**) Distribution of traction cables for each joint. (**c**) The screw-rod module containing three screw rods moves in a ratio of 1 : 2 : 3 through the gear set.

To highlight the novelty of underactuated SAMs, we compared the snake arm parameters produced by OC robotics and SIASUN robotics, as shown in Table 1. The following advantages can be achieved: (1) Under the premise of ensuring high spatial curvature and load capacity, the complexity of the drive system and control system is reduced by multiples. (2) The synchronous movement of adjacent joints considerably simplifies the kinematics model, and the inverse kinematics solution is more convenient and quicker. (3) A precise kinematics model can be established with high position accuracy. (4) The structural design can achieve miniaturization and lightweight, which are highly suitable

for the application requirements of complex and narrow spaces, such as those involving nuclear power [33].

**Table 1.** Comparison of the parameters of rigid hyper-redundant robots and SAMs [11,34].

| Item | OC Robotics | SIASUN | SAM |
|---|---|---|---|
| Total arm length | >2000 mm | 2269 mm | 2300 mm |
| Diameter | 140 mm | 125 mm | 80 mm |
| Number of joints | 12 | 12 | 10 |
| Number of motors | 36 + 1 | 36 + 1 | 12 + 1 |
| Degrees of freedom | 24 + 1 | 24 + 1 | 8 + 1 |
| Self-weight | >1000 kg | 1400 kg | <200 kg |
| Single bending angle | 27.5° | 22° | 25° |
| Maximum bending angle | 225° | 180° | 250° |
| Load | 10 kg | 5 kg | >5 kg |
| Length, width, and height | / | $3800 \times 800 \times 1550$ | $2900 \times 400 \times 500$ |
| Repeatable positioning accuracy | ±1 mm | ±1 mm | ±5 mm |

*2.2. SAM Kinematics Analysis*

There is a strong coupling relationship between the joint movements of the SAM, the cable displacement, and the rotation angle of the servo motor. The kinematic transformation of the robot can be decomposed into two parts to simplify the attendant calculation: the mapping of the operation space and the joint space; and the mapping of the joint space and the servo motor rotation angle. For a SAM composed of multiple universal joints in series, as shown in Figure 2, the kinematics model is relatively simple. The mapping relationship between the joint and operating spaces can be established using the classic D–H parameter method, as shown in Table 2. The homogeneous transformation matrix of adjacent universal joints in the joint group is as follows:

$$D_{2i+1}^{2i-1} = Tr(0, l_i, 0)R\left(y_{2i-1}, \frac{\pi}{2}\right)R(z_{2i}, \gamma_{2i})R\left(y_{2i}, -\frac{\pi}{2}\right)R(z_{2i+1}, \delta_{2i+1}) = \begin{bmatrix} c\delta_{2i+1} & -s\delta_{2i+1} & 0 & 0 \\ c\gamma_{2i}s\delta_{2i+1} & c\gamma_{2i}c\delta_{2i+1} & -s\gamma_{2i} & l_i \\ s\gamma_{2i}s\delta_{2i+1} & s\gamma_{2i}c\delta_{2i+1} & c\gamma_{2i} & 0 \\ 0 & 0 & 0 & 1 \end{bmatrix} \quad (1)$$

where $i$ takes the value 1, 2, 3 . . . ; $Tr$ is the translational transformation; $R$ is the rotational transformation; $c$ denotes $cos$; and $s$ denotes $sin$. Because the grouping drive is used, this gives $\delta_1 = \delta_3 = \delta_5 = \delta_a$, $\gamma_2 = \gamma_4 = \gamma_6 = \gamma_a$, and the angles of the remaining joint groups can be obtained by analogy. From this, the relationship between the SAM's end position and posture and the base coordinate system can be obtained as follows:

$$D_{21} = D_3^1 D_5^3 D_7^5 \cdots D_{21}^{19} \quad (2)$$

**Table 2.** SAM coordinate transformation D−H parameters.

| $i$ | $(\delta_i, \gamma_i)$ | Angle (°) | $l_i$ (mm) | Range (°) |
|---|---|---|---|---|
| 1 | $z_1(\delta_a)$ | (0, 90°, 0) | (0, 300, 0) | −25°~+25° |
| 2 | $z_2(\gamma_a)$ | (0, −90°, 0) | (0, 0, 0) | −25°~+25° |
| ⋮ | ⋮ | ⋮ | ⋮ | ⋮ |
| 19 | $z_{19}(\delta_d)$ | (0, 90°, 0) | (0, 200, 0) | −25°~+25° |
| 20 | $z_{20}(\gamma_d)$ | (0, −90°, 0) | (0, 0, 0) | −25°~+25° |

## 3. Methods

In this section, we use the equilibrium method and the forcibly equivalent joint torque method for the static and dynamic calculations of the SAM. Each joint of the SAM is drawn by three cables evenly distributed along the circumference to achieve pitch and yaw movements. This pertains to the problem of redundant drive, and the cable can only

provide positive tension. Therefore, the following assumptions are made regarding the dynamic model of the SAM: (1) The minimum cable preload force of any joint is set to 10 N in order to ensure that the cable always remains tight and to avoid the singularity problem of reverse thrust. (2) The frictional and environmental interferences are ignored. (3) The cable is simplified into a thin filament with ignorable mass, and the tension of the same cable is the same everywhere. (4) The joint mass is evenly distributed, along with the link.

### 3.1. Static Cable Force

To summarize the traction laws of SAMs, the force balance and torque balance methods can be used to analyze the cable traction characteristics of each joint of the snake arm along the horizontal direction of the $Y$-axis. As shown in Figure 4, we took the $i$-th joint as an example for static force analysis and decoupling calculations. This joint involves a total of seven external forces, where $F_1^i, F_2^i, F_3^i$ are the cable traction force; $F_4^i$ is the supporting force of the joint; $F_4^{i+1}$ is the reverse supporting force of the next joint; and $G_1^i, G_2^i$ are the gravity of the joint and the universal joint, respectively. Joint $i$ is also in a state of force balance and torque balance.

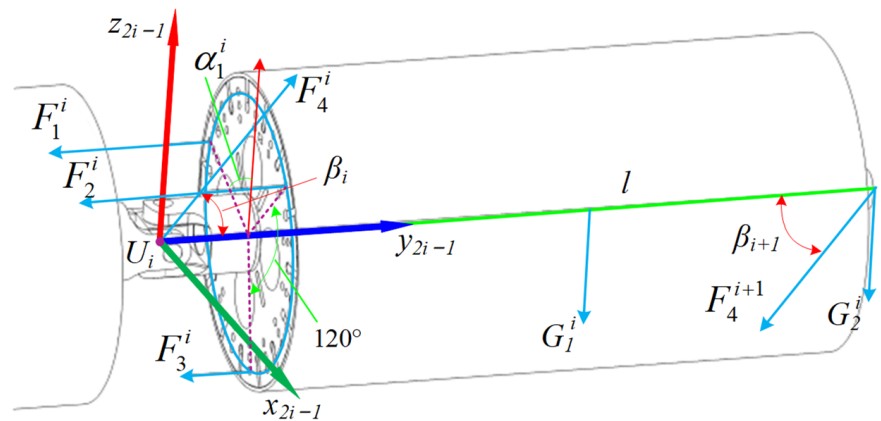

**Figure 4.** Force analysis of any joint $i$ balance state.

Torque balance analysis of $F_1^i, F_2^i, F_3^i$ indicates that $F_3^i$ is in a state of reaction torque. To avoid $F_3^i$ being negative, set $F_3^i = 10$ N. From the torque balance around the Z-axis and X-axis, the following can be obtained:

$$F_1^i r_i s\alpha_1^i + F_3^i r_i s\alpha_3^i + F_2^i r_i s\alpha_2^i = 0 \tag{3}$$

$$F_1^i r_i c\alpha_1^i + F_2^i r_i c\alpha_2^i + F_3^i r_i c\alpha_3^i + l\left(G_2^i + F_4^{i+1} s\beta_{i+1} + \frac{G_1^i}{2}\right) = 0 \tag{4}$$

where $r_i$ is the distance from the cable-fixing hole to the axis of the joint, $\alpha_1^i$ is the rotation angle of the first cable-fixing hole around the axis of the joint, and $\alpha_2^i = \alpha_1^i + \frac{2\pi}{3}$, $\alpha_3^i = \alpha_1^i - \frac{2\pi}{3}$. $\beta_i$ is the angle between the joint reaction force and the axis of the joint. Therefore:

$$F_1^i = \frac{F_3^i r_i \left(s\alpha_2^i c\alpha_3^i - c\alpha_2^i s\alpha_3^i\right) + \left(G_2^i + F_4^{i+1} s\beta_{i+1} + \frac{G_1^i}{2}\right) l s\alpha_2^i}{r_i \left(s\alpha_2^i c\alpha_1^i - c\alpha_2^i s\alpha_1^i\right)} \tag{5}$$

$$F_2^i = \frac{F_3^i r_i \left(s\alpha_1^i c\alpha_3^i - c\alpha_1^i s\alpha_3^i\right) + \left(G_2^i + F_4^{i+1} s\beta_{i+1} + \frac{G_1^i}{2}\right) l s\alpha_1^i}{r_i \left(s\alpha_2^i c\alpha_1^i - c\alpha_2^i s\alpha_1^i\right)} \tag{6}$$

From the force balance relationship, the following is obtained:

$$F_4^i = \sqrt{\left(F_1^i + F_2^i + F_3^i + F_4^{i+1}c\beta_{i+1}\right)^2 + \left(G_1^i + G_2^i + F_4^{i+1}s\beta_{i+1}\right)^2} \tag{7}$$

$$\beta_i = atan\frac{G_1^i + G_2^i + F_4^{i+1}s\beta_{i+1}}{F_1 + F_2 + F_3 + F_4^{i+1}c\beta_{i+1}} \tag{8}$$

Set the initial pose of the SAM as horizontally forward along the $Y$-axis, and set the end load $G$ to 0 N. Using the above algorithm to solve the cable traction force of each joint in the initial state, the cable traction force distribution map is obtained, as shown in Figure 5.

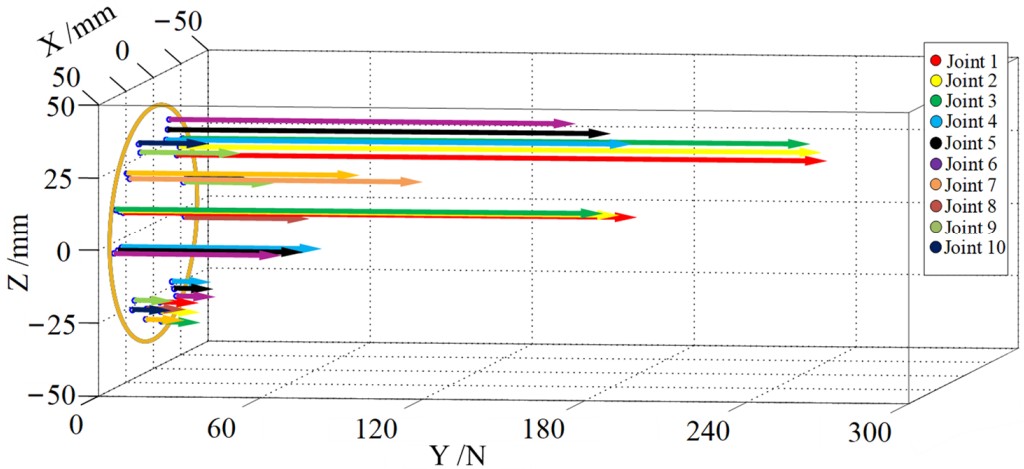

**Figure 5.** Traction force distribution of each cable in the initial pose.

## 3.2. Equivalent Joint Dynamics

The equivalent joint dynamics formed by the decoupling of the cable traction can control the SAM's movement. The cable traction forces of each joint of the SAM are dynamically coupled. Moreover, each joint involves some acceleration, meaning the external force and external torque received are also unbalanced. Solving the cable traction force using traditional force analysis and dynamic equations is highly complicated. In this paper, we propose an equivalent dynamics modelling method for the strongly coupled dynamic characteristics of each joint cable. The effects of all joint cable traction forces are forcibly equated to joint torques to create a dynamic model. Then, the joint torque can be converted into the traction force of the cable through the forcibly equivalent inverse transformation, and the real-time decoupling calculation of the cable traction force can be realized.

First, the equivalent joint dynamics of the end joint of the SAM are decoupled in any given pose. As shown in Figure 6, using force analysis, the equivalent transformation of the cable traction force $(F_1^{10}, F_2^{10}, F_3^{10})$ and the joint driving torque $(T_1^{10}, T_2^{10})$ can be realized. Then, the forcibly equivalent SAM dynamics equations can be obtained by traditional dynamics modelling methods (Lagrangian, Newtonian–Euler, etc.). As shown in Figure 6, assume that the equivalent joint torques $(T_1^{10}, T_2^{10})$ are known, and solve for the cable traction force $(F_1^{10}, F_2^{10}, F_3^{10})$. First, solve the direction vectors of $F_1^{10}, F_2^{10}, F_3^{10}$ and decompose them to the coordinate systems $(x_2, y_2, z_2)$ and $(x_3, y_3, z_3)$ of point $A_2$. Then, we solve the value of $F_1^{10}, F_2^{10}, F_3^{10}$ via the forcibly equivalent method. Because a single joint is controlled by three cables and has the characteristic of redundant driving, countless solutions exist. To limit the number of solutions and avoid reverse tension, set the minimum cable preload force to 10 N. By analyzing the cable layout of the end joints, the following is obtained: $F_3^{10} = 10$ N.

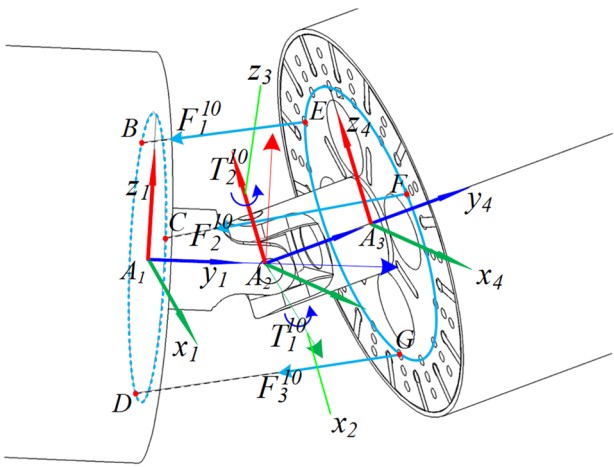

**Figure 6.** Force analysis of end joints in any posture.

The following could be used to solve the direction vectors of $F_1^{10}, F_2^{10}, F_3^{10}$ in the coordinate system: $(x_2, y_2, z_2)$ and $(x_3, y_3, z_3)$ of point $A_2$. The coordinates of point $B$ and $E$ in the coordinate system $(x_2, y_2, z_2)$ are:

$$B(x_{B2}, y_{B2}, z_{B2}) = \textbf{\textit{Tr}}(0, -h, 0) \begin{bmatrix} -r sin\alpha_1^{10} \\ 0 \\ r cos\alpha_1^{10} \\ 1 \end{bmatrix} \tag{9}$$

$$E(x_{E2}, y_{E2}, z_{E2}) = \textbf{\textit{R}}(x_2, \gamma_d)\textbf{\textit{R}}(z_3, \delta_d)\textbf{\textit{Tr}}(0, h, 0) \begin{bmatrix} -r sin\alpha_1^{10} \\ 0 \\ r cos\alpha_1^{10} \\ 1 \end{bmatrix} \tag{10}$$

where $2h$ is the length of the universal joint, $\gamma_d$ is the rotation angle of the universal joint around the $x_2$ axis, $\delta_d$ is the rotation angle of the universal joint around the $z_3$ axis, and $\alpha_1^{10}$ is the rotation angle of the cable hole around the axis of the joint.

The unit direction vector of $F_1^{10}$ in the coordinate system $(x_2, y_2, z_2)$ can be obtained as follows:

$$\overrightarrow{EB} = \frac{(x_{B2} - x_{E2},\ y_{B2} - y_{E2},\ z_{B2} - z_{E2})}{\sqrt{(x_{B2} - x_{E2})^2 + (y_{B2} - y_{E2})^2 + (z_{B2} - z_{E2})^2}} \tag{11}$$

Then, decompose $F_1^{10}$ into the parallel lines of each axis of the coordinate system $A_2(x_2, y_2, z_2)$ according to the force vector decomposition method:

$$F_{x2F1}^{10} = \frac{(x_{B2} - x_{E2})}{\sqrt{(x_{B2} - x_{E2})^2 + (y_{B2} - y_{E2})^2 + (z_{B2} - z_{E2})^2}} F_1^{10} \tag{12}$$

$$F_{y2F1}^{10} = \frac{(y_{B2} - y_{E2})}{\sqrt{(x_{B2} - x_{E2})^2 + (y_{B2} - y_{E2})^2 + (z_{B2} - z_{E2})^2}} F_1^{10} \tag{13}$$

$$F_{z2F1}^{10} = \frac{(z_{B2} - z_{E2})}{\sqrt{(x_{B2} - x_{E2})^2 + (y_{B2} - y_{E2})^2 + (z_{B2} - z_{E2})^2}} F_1^{10} \tag{14}$$

Similarly, $F_2^{10}, F_3^{10}$ can be decomposed to the parallel lines of each axis of the coordinate system $A_2(x_2, y_2, z_2)$ according to the vector decomposition method of force, with the following results: $F_{x2F2}^{10}, F_{y2F2}^{10}, F_{z2F2}^{10}$ and $F_{x2F3}^{10}, F_{y2F3}^{10}, F_{z2F3}^{10}$.

Therefore, the torques produced by $F_1^{10}, F_2^{10}, F_3^{10}$ on the $x_2$ axis of the coordinate system of point $A_2(x_2, y_2, z_2)$ can be obtained as follows:

$$T_{x2F1}^{10} = F_{y2F1}^{10} z_{E2} + F_{z2F1}^{10} y_{E2} \tag{15}$$

$$T_{x2F2}^{10} = F_{y2F2}^{10} z_{F2} + F_{z2F2}^{10} y_{F2} \tag{16}$$

$$T_{x2F3}^{10} = F_{y2F3}^{10} z_{G2} + F_{z2F3}^{10} y_{G2} \tag{17}$$

From the forcibly equivalent transformation, the following can be obtained:

$$\mathbf{T_1^{10}} = T_{x2F1}^{10} + T_{x2F2}^{10} + T_{x2F3}^{10} \tag{18}$$

Similarly, the forcibly equivalent transformation of $\mathbf{T_2^{10}}$ on the $z_3$ axis in the coordinate system $A_2(x_3, y_3, z_3)$ can be used to obtain:

$$\mathbf{T_2^{10}} = T_{z3F1}^{10} + T_{z3F2}^{10} + T_{z3F3}^{10} \tag{19}$$

Upon setting the minimum traction force to $\mathbf{F_3^{10}} = 10$ N, Equation (18) can be combined with Equation (19) to solve real-time $\mathbf{F_1^{10}}$ and $\mathbf{F_2^{10}}$.

The force characteristics of any joint (except the end joint) of the SAM are more complicated than those of the end joint. In addition to the cable traction force of the joint itself, all cables after the joint have forces acting on them. As shown in Figure 7, we took the $i$-th joint ($i < 10$) as an example for the force analysis and decoupling calculations. Compared with the end joint shown in Figure 6, the $i$-th joint has its own cable traction forces ($F_1^i, F_2^i, F_3^i$), in addition to the cable traction forces from the $(i + 1)$-th, ... , the 10th joints (end joints). As shown in Figure 7a, $F_{j1}^i, F_{j2}^i, F_{j3}^i$ are the forces generated by the cables of the $j$-th joint on the midline of the cable clamping angle ($i < j \leq 10$).

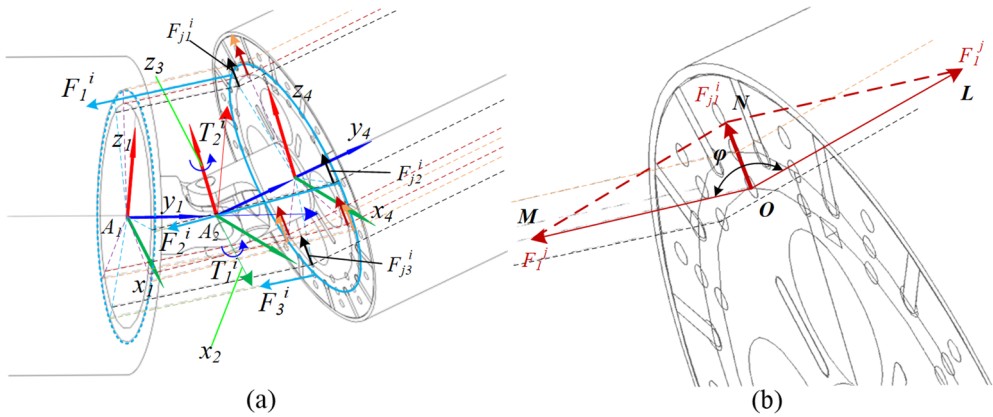

(a)  (b)

**Figure 7.** Force analysis of any joint in any posture (the $i$-th joint was taken as an example). (**a**) $i$-joint coordinate system and cable forces. (**b**) Local enlargement of $F_1^j$.

For the cable traction force ($F_1^i, F_2^i, F_3^i$) of the $i$-th joint shown in Figure 7a, the solution algorithm is similar to that of the end joint and will not be repeated here; rather, we focus on analyzing the influence of the traction force of the $j$-th joints ($i < j \leq 10$) on the $i$-th joint. In Figure 7b, the cable traction force ($F_1^j$) of the $j$-th joint is drawn as a spatial parallelogram $(O, L, N, M)$, and its diagonal $(ON)$ is $F_{j1}^i$, which is the force acting on the $i$-th joint.

Using coordinate transformation, the $O, L, N, M$ points are respectively transformed into the coordinate system $(x_2, y_2, z_2)$ and $(x_3, y_3, z_3)$, with $A_2$ as the coordinate origin. In the $A_2(x_2, y_2, z_2)$ coordinate system, the coordinates of point $O(x_{O2}, y_{O2}, z_{O2})$, as well as the size and direction of the vector ($F_1^j$) are known. The coordinates of $L(x_{L2}, y_{L2}, z_{L2})$ and $M(x_{M2}, y_{M2}, z_{M2})$ can be obtained using matrix transformation and the distance formula

between two points. From the geometric properties of the parallelogram in space, the coordinates of $N(x_{N2}, y_{N2}, z_{N2})$ can be obtained as follows:

$$\begin{cases} x_{N2} = (x_{M2} - x_{O2}) + x_{L2} \\ y_{N2} = (y_{M2} - y_{O2}) + y_{L2} \\ z_{N2} = (z_{M2} - z_{O2}) + z_{L2} \end{cases} \tag{20}$$

From this, the unit direction vector ($\overrightarrow{ON}$ of $F_{j1}^i$) can be obtained as follows:

$$\overrightarrow{ON} = \frac{(x_{N2} - x_{O2},\ y_{N2} - y_{O2},\ z_{N2} - z_{O2})}{\sqrt{(x_{N2} - x_{O2})^2 + (y_{N2} - y_{O2})^2 + (z_{N2} - z_{O2})^2}} \tag{21}$$

Meanwhile, from the calculation formula of the angle between two non-zero vectors:

$$\cos(\varphi) = \frac{\overrightarrow{OM}\ \overrightarrow{OL}}{|OM||OL|} \tag{22}$$

It is clear that the size of $F_{j1}^i$ is as follows:

$$F_{j1}^i = 2F_1^j \cos\left(\frac{\varphi}{2}\right) \tag{23}$$

According to the force vector decomposition method, $F_{j1}^i$ is decomposed to $A_2(x_2, y_2, z_2)$ on the parallel lines of each axis:

$$F_{x2Fj1}^i = \frac{(x_{N2} - x_{O2})}{\sqrt{(x_{N2} - x_{O2})^2 + (y_{N2} - y_{O2})^2 + (z_{N2} - z_{O2})^2}} F_{j1}^i \tag{24}$$

$$F_{y2Fj1}^i = \frac{(y_{N2} - y_{O2})}{\sqrt{(x_{N2} - x_{O2})^2 + (y_{N2} - y_{O2})^2 + (z_{N2} - z_{O2})^2}} F_{j1}^i \tag{25}$$

$$F_{z2Fj1}^i = \frac{(z_{N2} - z_{O2})}{\sqrt{(x_{N2} - x_{O2})^2 + (y_{N2} - y_{O2})^2 + (z_{N2} - z_{O2})^2}} F_{j1}^i \tag{26}$$

The torque produced by $F_{j1}^i$ on the $x_2$ axis at $A_2(x_2, y_2, z_2)$ can be obtained as follows:

$$T_{x2Fj1}^i = F_{y2Fj1}^i z_O + F_{z2Fj1}^i y_O \tag{27}$$

In the same way, close to $F_2^j, F_3^j$, there are also cable forces of $F_{j2}^i$ and $F_{j3}^i$. Accordingly, the torques generated on the $x_2$ axis of the $A_2(x_2, y_2, z_2)$ coordinate system are $T_{x2Fj2}^j$ and $T_{x2Fj3}^j$.

From the forcibly equivalent transformation of $T_1^i$ on the $x_2$ axis of $A_2(x_2, y_2, z_2)$ and $T_2^i$ on the $z_3$ axis of $A_2(x_3, y_3, z_3)$, the following can be obtained:

$$T_1^i = T_{x2F1}^i + T_{x2F2}^i + T_{x2F3}^i + T_{x2Fj1}^i + T_{x2Fj2}^i + T_{x2Fj3}^i + \cdots \tag{28}$$

$$T_2^i = T_{z3F1}^i + T_{z3F2}^i + T_{z3F3}^i + T_{z3Fj1}^i + T_{z3Fj2}^i + T_{z3Fj3}^i + \cdots \tag{29}$$

Upon setting the minimum traction force to $F_3^i = 10$ N, Equation (28) can be combined with Equation (29) to solve $F_1^i, F_2^i$. In the same way, the cable traction force of any joint can be obtained.

For $T_1^i, T_2^i$, the Lagrangian equations can be used to establish equivalent dynamic equations for the calculation [35,36]. The cable traction is equivalent to the joint torque, and

the SAM becomes a typical serial multijoint manipulator, with the final dynamic model expressed as follows:

$$T_d = M(q)\ddot{q} + H(q,\dot{q})\dot{q} + G(q) + D \tag{30}$$

where $\ddot{q} \in R^{20}$ is the joint angular acceleration vector, $\dot{q} \in R^{20}$ is the joint velocity vector, $T \in R^{20}$ is the input torque vector, $M(q) \in R^{20 \times 20}$ is a nonsingular positive definite inertial force matrix, $H(q,\dot{q}) \in R^{20 \times 20}$ is the term of centrifugal force and Coriolis force, $G(q)$ is the gravitational term (including the gravity of the connecting rod, the universal joint, the end tools, and the load), and $D$ represents the unknown bounded disturbance of the unstructured, unbuilt dynamic model.

## 4. Model-Based Control

Traditional $PID$ control is widely used in industry due to its simplicity and reliability. However, when the system dynamic model is complex and variable, simple $PID$ control cannot provide sufficient accuracy. Model-based control can effectively improve the robustness of the system. Therefore, the validity of the equivalent dynamics can be verified by introducing a simple dynamics model-based $PID$ controller [37,38].

As shown in Figure 8, in the working process of the snake robot, the first step is to perform online or offline trajectory planning to obtain the inverse kinematics solution $(\delta_i, \gamma_i)$ of each joint group. Then, the kinematics inverse solution is divided into two channels: one channel is transformed into the motor-driving signal $(\theta_i)$. Another channel imports the Lagrangian dynamic model to solve the driving torque of each joint and obtain the feedforward torque $(T_{id})$ of the motor through equivalent transformation and decoupling calculation. The model-based feedforward torque $(T_{id})$ of the motor plus the motor drive torque $(T_{ik})$ after $PID$ tuning jointly control the output torque $(T_i)$ of the motor, as well as the movement of the robot. The kinematic data $(\theta_{ip})$ of the robot movement process are collected and fed back to the closed-loop controller.

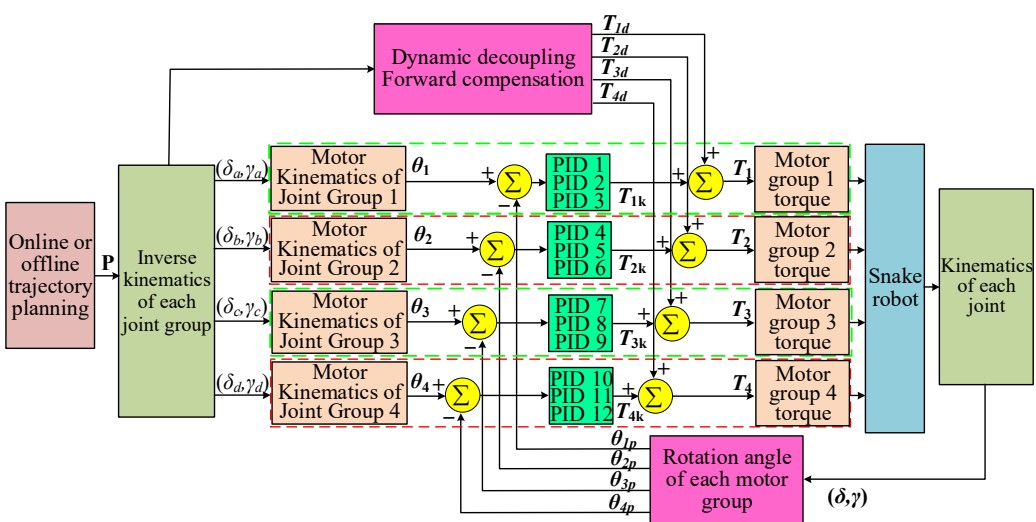

**Figure 8.** Model-based equivalent dynamic control principle.

The torque $(T)$ of each joint is expressed as follows:

$$T = T_d + T_k \tag{31}$$

$$T_k = k_p e(t) + k_i \sum e(t) + k_d \dot{e}(t) \tag{32}$$

where $e(t) = \theta(t) - \theta_p(t)$ is the motor-angle tracking deviation; $k_p$ is the proportional coefficient, which responds to the current deviation of the system ($e(t)$); $k_i$ is the integral coefficient, which responds to the accumulated deviation of the system ($\sum e(t)$); and $k_d$ is the differential coefficient, which reflects the rate of change of the system deviation signal

($\dot{e}(t)$). By combining the equivalent dynamics model ($T_d$) of SAM and the compensation output ($T_k$), the joint control torque ($T$) can be obtained.

## 5. Simulation and Experiments

### 5.1. Simulation Tests

To further verify the effectiveness of the equivalent dynamics control, a snake robot with an arm length of 1500 mm (including 6 joints) was simulated and analyzed using simulation software, as shown in Figure 9. Set the target curve for each joint of the snake robot to rotate around each axis, as shown in Figure 10a,c. The first joint group moves in a sinusoidal curve, and the second joint group remains stationary. Motion control is carried out by the model-based equivalent dynamic controller. The tracking errors of each joint are shown in Figure 10b,d, and the stability error is less than $1 \times 10^{-3}$ rad. The simulation results verify the validity of the equivalent dynamic model and the feasibility of its practical application.

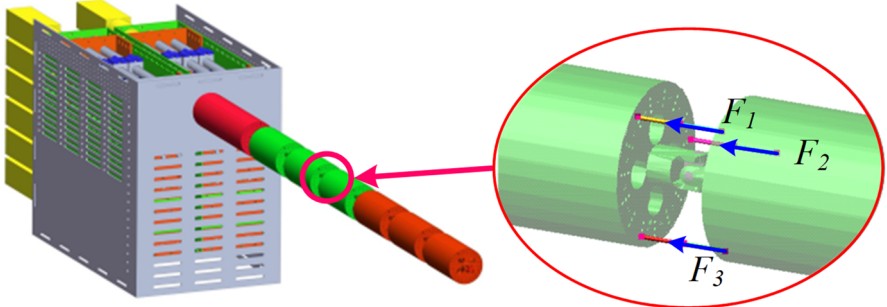

**Figure 9.** Simulation analysis of snake robot equivalent dynamic control.

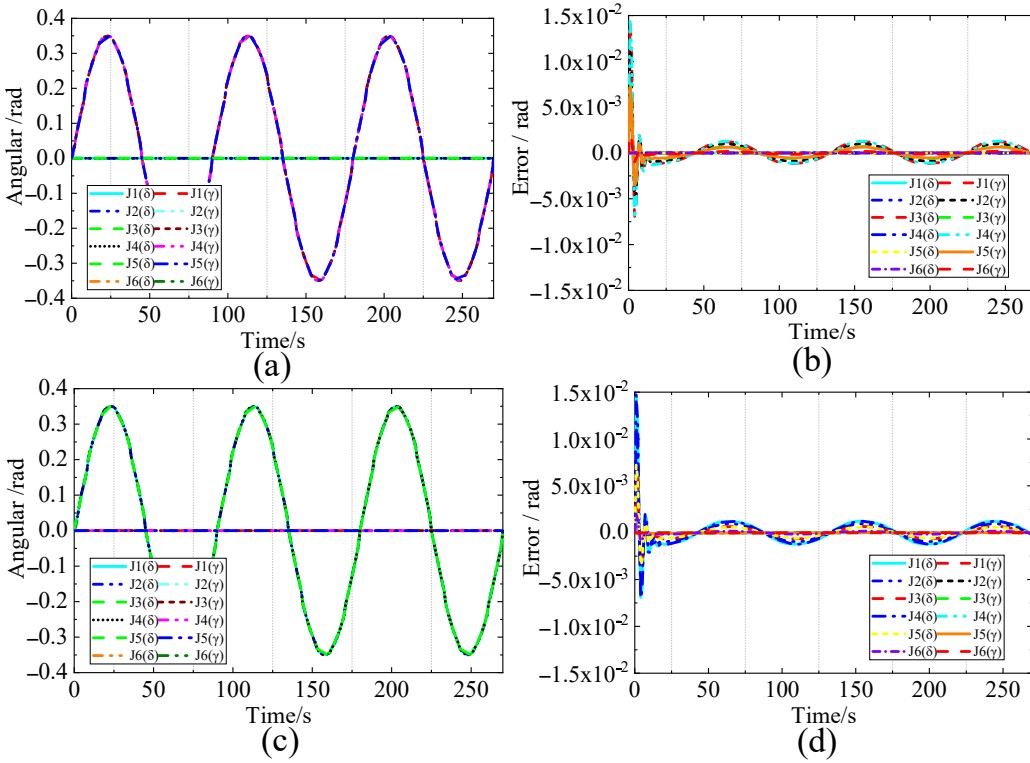

**Figure 10.** Simulation analysis of the controller. (**a**,**b**) Vertical sinusoidal trajectory and error and (**c**,**d**) horizontal sinusoidal trajectory and error.

### 5.2. Traction Force Experiments

The effectiveness of the above equivalent dynamic control in engineering applications depends on the accuracy of the theoretical equivalent dynamics model. The higher the accuracy of the equivalent dynamics, the better the control of the controller. Therefore, we designed a SAM prototype, as shown in Figure 11, for experimental comparison of the accuracy of the theoretical equivalent dynamics model. The dimensions of each joint and drive box are $80 \times 80 \times 200$ mm and $400 \times 300 \times 430$ mm, respectively. The motor uses a 600 W AC servo motor with its own power-off brake and encoder, whereas the motion controller uses a high-end multiaxis motion control card.

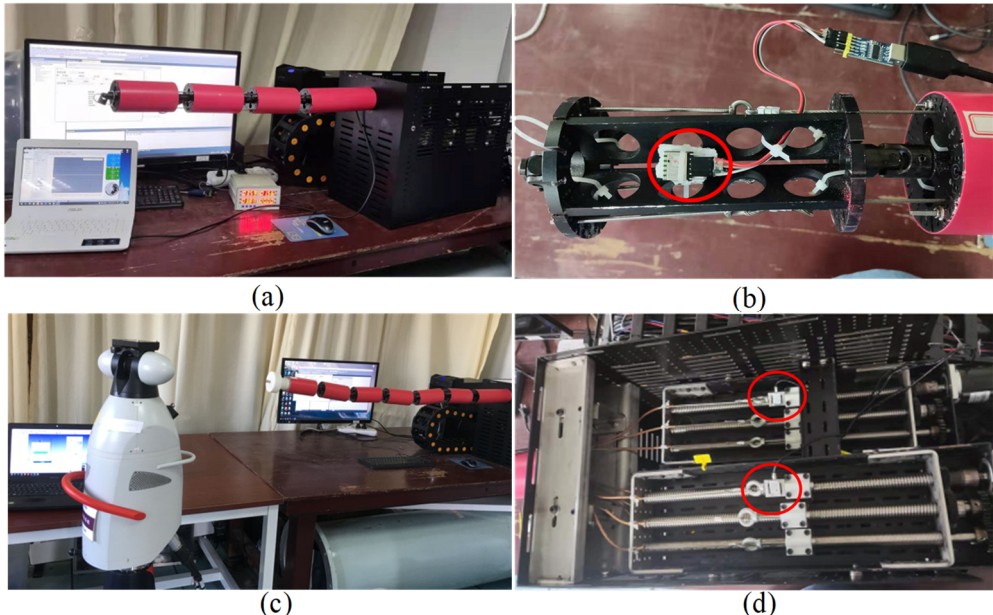

**Figure 11.** SAM prototype test. (**a**) Single-joint group prototype. (**b**) Gyro angle sensor installation position. (**c**) Laser tracker end position calibration. (**d**) Force sensor installation position.

Two SAMs with different arm lengths were designed to compare and verify the dynamics and positional accuracy. See Supplementary Materials for videos of the experiments. Figure 11a,b shows the single-joint group test prototype of the SAM. Here, the single-joint group contains three rigid joints, and the drive cables of each joint are distributed according to a specific position, as shown in Figure 11b. The rotation angle of each joint in the joint group was measured by installing a gyroscope angle sensor in the foremost joint unit. The driving force of each cable in the joint group was measured by installing a tension sensor in the end drive box, as shown in Figure 11d. A calibration test of the end position accuracy of the SAM using a laser tracker is shown in Figure 11c.

The structural rationality and dynamic characteristics were analyzed using the open-loop speed prospective control test. Here, the joint group was set to complete three cycles of sine motion trajectory in both the horizontal and vertical directions. The end-effector position of the SAM is first calibrated using a laser tracker. The angle sensor data of the end joint are shown in Figure 12a,c. The joints in the joint group can track the target trajectory effectively during the movement, with an average angle error of less than $1 \times 10^{-2}$ rad, which verifies the rationality of the underactuated continuum joint design. The cable traction forces of the end joint measured by the force sensor are shown in Figure 12b,d. During each joint angular velocity zero and reverse acceleration phase, a large change in cable traction force occurs. This is mainly caused by the increased friction of the cable in the joint through-hole. Next, a forcibly equivalent dynamic transformation of the experimentally obtained cable traction forces is performed based on the decoupling algorithm described in Section 3.

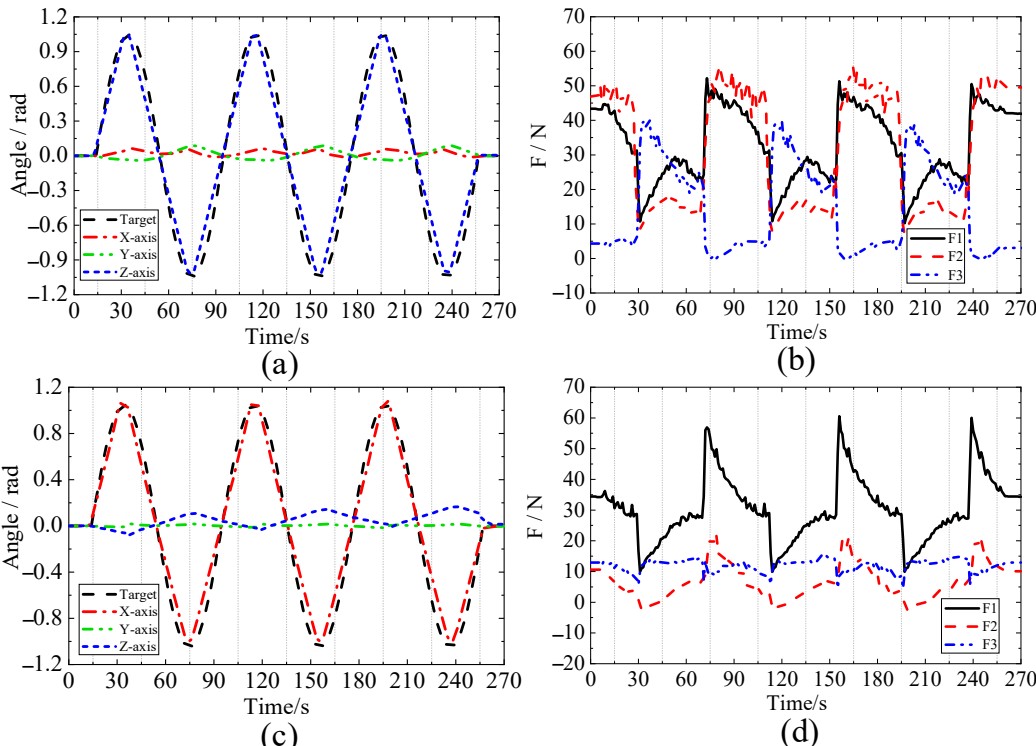

**Figure 12.** Test results of the single-joint group. (**a**,**b**) Vertical sinusoidal trajectory and cable force and (**c**,**d**) horizontal sinusoidal trajectory and cable force.

### 5.3. Equivalent Joint Dynamic Analysis

The joint dynamics are obtained by equivalently transforming the SAM cable traction forces using the decoupling algorithm. To verify the accuracy of the decoupling algorithm, we also calculated the theoretical joint torques of the end joints under the same trajectory based on Adams dynamics software. The dynamics of the flexible cables are calculated by equating them to the linear springs with the addition of linear damping constraints. The relationship between the total equivalent joint torques and the theoretical joint torques calculated by Adams software are shown in Figure 13a,b. According to comparison of the results of the decoupling of the cable traction forces with the theoretical results of the joint torques, the forcibly equivalent joint torques obtained from the decoupling experiments have the same movement trend as the theoretical joint torques obtained from the Adams software. This proves that the decoupling algorithm for cable traction is effective.

Figure 13c,d shows the error characteristics of the theoretical torques in relation to the equivalent joint torques. The maximum torque error is close to 0.38 Nm, but the mean error and standard deviation are small (less than 0.13 Nm). The large errors mainly reflect theoretically unbuilt model errors, such as the rigid–flexible coupling deformation of the cables and non-linear friction. In general, the SAM equivalent joint torques have the same motion trend as the Adams theoretical torques, and the data in Figure 13 show that the difference between the theoretical joint torques and the experimental equivalent joint torques is significantly smaller than the experimentally equivalent joint torques. Therefore, equivalent dynamic control can improve the control effect. The experimental results can also guide the development of more accurate theoretical dynamics models.

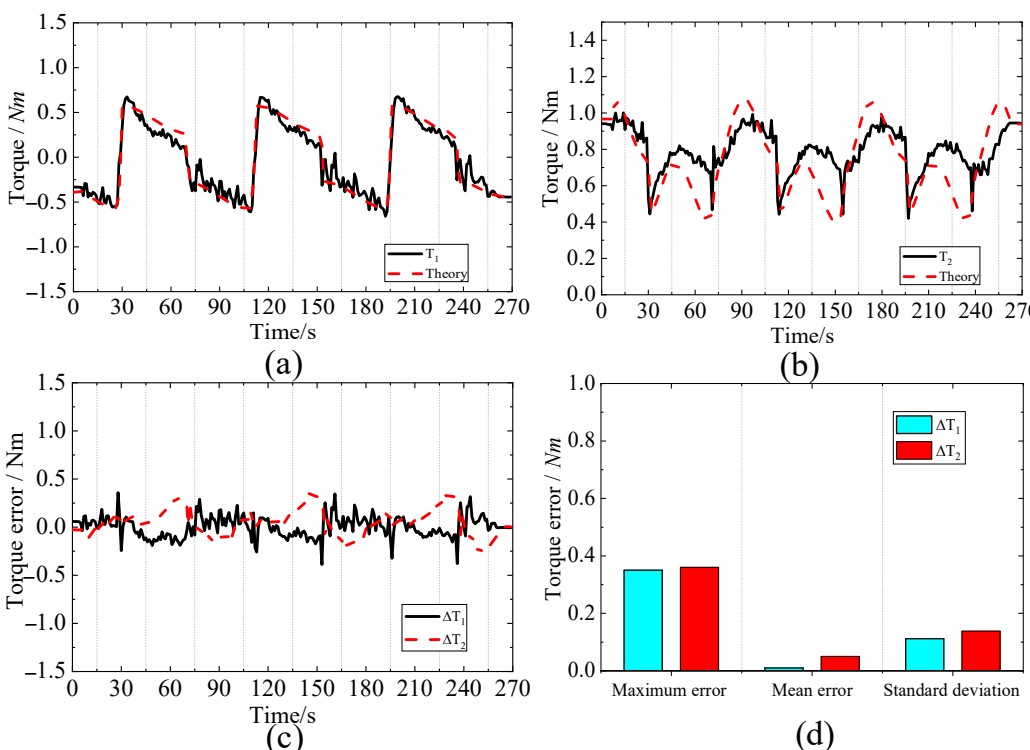

**Figure 13.** Comparative analysis of theoretical and experimental data of equivalent joint torques. (**a**) Theoretical and experimental data of $T_1$ under vertical sinusoidal trajectory. (**b**) Theoretical and experimental data of $T_2$ under horizontal sinusoidal trajectory. (**c**) $T_1$ and $T_2$ errors. (**d**) Analysis of the error characteristics of $T_1$ and $T_2$.

## 6. Conclusions

In this paper, an underactuated SAM was designed, and the attendant kinematics were analyzed based on the application requirements of the narrow spaces of a nuclear power plant. Based on various force balance and torque balance algorithms, the static characteristics of each joint of the SAM in the initial pose were analyzed. Based on the principle of equivalent transformation and the Lagrangian equation, a decoupling algorithm for the strong coupling dynamics of the SAM was proposed. The traction force of the cable coupled between the joints was equivalent to the joint torque, and the real-time equivalent joint torque was obtained. The decoupling calculation of the cable traction was then realized via inverse transformation of the joint torque. To verify the correctness of the equivalent dynamics, we designed a model-based dynamic controller based on the equivalent dynamics. The simulation results demonstrate the effectiveness of SAM motion control based on equivalent dynamics. The maximum tracking error at a steady state is less than $1 \times 10^{-3}$ rad. In addition, a SAM prototype with a single-joint group and a 1500 mm arm length was created, and experimental data were collected. Accuracy tests on a 1500 mm arm-length SAM validate the rationality of the underactuated design. Using the open-loop control trajectory control experiment of the SAM single-joint group, the cable force, joint angle tracking error, and equivalent joint torques were analyzed to verify the accuracy of the equivalent dynamic model.

The theoretical dynamic model will be further optimized in future studies. The main areas to be optimized are as follows: (1) The equivalent friction model of the cable in the circular hole will be developed to identify non-linear friction. (2) The linear elastic deformation model of the cable will be developed to identify the dynamics of flexible errors. (3) Neural network and machine learning will be used to identify and compensate for the unknown dynamic parameters.

**Supplementary Materials:** The following supporting information can be downloaded at: https://www.youtube.com/channel/UCOJ1DxVz8Qa1TlO5XOKpuGg, Video S1: Left and right.mp4, Video S2: Up and down.mp4 (accessed on 14 July 2022). And https://www.mdpi.com/article/10.3390/app12157494/s1, File S1: Supplementary materials.docx.

**Author Contributions:** G.Q.: methodology, validation, software, data curation, writing—original draft; A.J.: conceptualization, formal analysis, funding acquisition, investigation; H.W.: validation, writing—review and editing, project administration. All authors have read and agreed to the published version of the manuscript.

**Funding:** This work was carried out within the framework of the EUROfusion Consortium, funded by the European Union via the Euratom Research and Training Programme (Grant Agreement No. 101052200—EUROfusion). Views and opinions expressed are those of the author(s) only and do not necessarily reflect those of the European Union or the European Commission. Neither the European Union nor the European Commission can be held responsible for them. This work is supported by the National Key R&D Program of China (Grant No. 2019YFB1309600), the National Natural Science Foundation of China (Grant Nos. 51875281 and 51861135306), and the China National Special Project for Magnetic Confinement Fusion Science Program (Grant No. 2017YFE0300503).

**Institutional Review Board Statement:** Not applicable.

**Informed Consent Statement:** Not applicable.

**Data Availability Statement:** Not applicable.

**Acknowledgments:** The authors are grateful to Huan Shen, Qian Li, Qingfei Han, Zhikang Yang, and Shikun Wen for their participation in valuable discussions at various stages of the work. The authors also thank AISPP Institute for participating in technical discussions. Finally, the authors would like to thank the editors and reviewers for their valuable comments and constructive suggestions.

**Conflicts of Interest:** The authors declare no conflict of interest.

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
