# Peer review of "Equivalent Dynamic Analysis of a Cable-Driven Snake Arm Maintainer"

_applsci, doi:10.3390/app12157494_

Round 1

Reviewer 1 Report

The author carried out an interesting study. The paper is novel, the research content is rich, there are complex and precise calculation, simulation and experiment, the workload is relatively full. There are some minor suggestions:

1. The innovation of the paper should be reflected in the abstract and introduction.

2. It is suggested to add more notes in Figure 1。

3. In the article, some formulas lack sources and some derivation is not detailed.

4. The data analysis section could be fleshed out.

5. The literature analysis in the introduction can be richer.  For example, you can cite some other equipment or equipment dynamics analysis articles. Such as "Research on the vibration model and vibration performance of cold orbital forging machines" In this paper, vibration characteristics are obtained by dynamic analysis of a manufacturing process, and experiments are carried out to verify the effectiveness of dynamic analysis. In addition, vibration reduction methods are also studied to reduce. This article is very valuable for reference and analysis.

As a whole, this is a good paper, my advice is minor revision.

Author Response

We are very grateful to the reviewers for their review of the manuscript. We have answered all the questions and have marked them in red in the manuscript. Please check. 

For the response to each question please see the attachment. 

Reviewer 2 Report

The paper presents a cable-driven snake arm maintainer and its analysis based on dynamic modeling. The authors derived kinematics and dynamics based on each joint force with constraints. The authors fabricated 10 joints snake robot and conducted experimental and simulation validation. Major concerns are;

1. Given the long history of the similar robot system, the novelty and contribution of the paper are not clearly shown. A comparison table with the previous mechanism will be helpful to emphasize these points.

2. Please add an explanation of why the proposed mechanism is redundant. Although the system has 10 joints, those joints are controlled by 6 cables-motors for each joint. Then, the number of actuators should be counted as more than 6.

3. What’s the form of M, H, G, D in equation (30)

4. The authors derived a bunch of dynamic models but control is achieved by the PID control method that is less utilizing the model and Fig. 13 only depicts T1 and T2 for analysis. Then, the necessity of the dynamic model derivation is not clearly shown. A dynamic model simulation for different robot shapes or postures will be helpful for qualified publication.

5. Experimental results of different posture/shape is requested to be added. 

Author Response

(The authors gave the same response as above.)

Round 2

Reviewer 2 Report

The paper is well revised for publication.